# BIAS RESILIENT MULTI-STEP OFF-POLICY GOAL-CONDITIONED REINFORCEMENT LEARNING

## ABSTRACT

In goal-conditioned reinforcement learning (GCRL), sparse rewards present significant challenges, often obstructing efficient learning. Although multi-step GCRL can boost this efficiency, it can also lead to off-policy biases in target values. This paper dives deep into these biases, categorizing them into two distinct categories: "shooting" and "shifting". Recognizing that certain behavior policies can hasten policy refinement, we present solutions designed to capitalize on the positive aspects of these biases while minimizing their drawbacks, enabling the use of larger step sizes to speed up GCRL. An empirical study demonstrates that our approach ensures a resilient and robust improvement, even in ten-step learning scenarios, leading to superior learning efficiency and performance that generally surpass the baseline and several state-of-the-art multi-step GCRL benchmarks.

## 1 INTRODUCTION

In goal-conditioned reinforcement learning (GCRL) (Schaul et al., 2015a; Liu et al., 2022), agents aim to complete various tasks each characterized by different goals. A primary challenge arises from the complexity of reward engineering, often leading to sparse reward settings—where valuable rewards are only given upon reaching the goals. Such limited feedback inhibits efficient learning, causing delays in the learning progression. Although relabeling goals based on actual outcomes for each trajectory augments the learning experience (Andrychowicz et al., 2017), the slow propagation of these sparse learning signals remains a bottleneck for the rapid acquisition of successful policies. To counteract this, multi-step hindsight experience replay (MHER) (Yang et al., 2021) accelerates the relay of relabeled rewards using multi-step targets. However, in the off-policy setting, multi-step learning can introduce off-policy biases originating from the divergence between target and behavior policies. As the step-size magnifies, these challenges are further amplified. While methods like MHER($\lambda$) and MMHER, proposed by Yang et al. (2021), attempt to mitigate these biases, we observe that they lack the requisite resilience, rendering their methods effective only for smaller step sizes.

Though off-policy bias in multi-step RL has been well-studied (Sutton & Barto, 2018; Munos et al., 2016; De Asis et al., 2018), its nuances within GCRL present novel challenges. In GCRL, the mere act of reaching a goal does not guarantee its sustained achievement. The agent must learn the optimal strategy to consistently stay on the goal, which necessitates continuous bootstrapping of value functions on goal states. As off-policy biases accumulate at these goal states, they can retroactively distort the value estimates of preceding states. In light of this, we categorize off-policy bias into the *shooting* bias that accumulates up to the goal states and introduce the novel concept of *shifting* bias that builds within the goal states. While off-policy biases are typically perceived as detrimental, by examining the multi-step target values, we propose that certain behavior policies offer superior alternatives over the multi-step reward accumulation, guiding the agent to refine its policy more swiftly. Building on this understanding, we present tailored solutions that harness the positive effects of both biases while curbing their adverse effects.

In the broader context of examining bias problems in multi-step GCRL, it is imperative to recognize that the overall bias is not confined to off-policy bias. It also includes over-optimistic bias (Hasselt, 2010; Van Hasselt et al., 2016; Fujimoto et al., 2018) and hindsight biases (Blier & Ollivier, 2021; Schramm et al., 2023). Therefore, to isolate the effects and better understand the intricate nature of off-policy bias, we utilize existing techniques (Fujimoto et al., 2018) to counteract the over-optimistic

bias and completely bypass hindsight biases stemming from stochastic dynamics (Blier & Ollivier, 2021; Schramm et al., 2023) by centering our study on deterministic environments.

Our main contributions are summarized as follows: i) By analyzing off-policy bias in multi-step GCRL, we dissect off-policy bias in multi-step GCRL, cagorizing it into two distinct types based on their respective roles, and introduce metrics for evaluating each type of bias. ii) We probe the root causes of two types of biases in multi-step GCRL, elucidating their beneficial and detrimental effects on learning. iii) We propose a novel resilient strategy for each type of off-policy bias in multi-step GCRL, culminating in the robust algorithm BR-MHER. iv) Empirical evaluation demonstrates that our approach robustly outperforms both the HER baseline and several state-of-the-art multi-step GCRL methods in efficiency, bias mitigation, and performance across varied GCRL benchmarks.

## 2 RELATED WORK

**Multi-Step Off-Policy RL**    In off-policy multi-step RL, addressing off-policy bias is paramount. There are several identified biases: overestimation bias stemming from function approximation errors (Hasselt, 2010; Van Hasselt et al., 2016; Fujimoto et al., 2018), distributional shift bias due to imbalanced learning distributions (Schaul et al., 2015b), and hindsight bias which manifests in stochastic environments (Blier & Ollivier, 2021; Schramm et al., 2023). Both TD($\lambda$) and Q($\lambda$) methods utilize eligibility traces as a means to mitigate such biases (Sutton & Barto, 2018; Harutyunyan et al., 2016). Additionally, Q($\lambda$) (Harutyunyan et al., 2016) brings in off-policy corrections using the current action-value functions. Importance Sampling (IS) offers an additional layer for bias correction and aids in convergence (Precup et al., 2001; Munos et al., 2016), but it does come with a trade-off of heightened variance. Despite these corrective measures, a majority of these methods tend to cap their step size, often limiting it to three steps (Barth-Maron et al., 2018; Hessel et al., 2018).

**Multi-Step Off-Policy GCRL**    Multi-step GCRL has inherent biases akin to traditional multi-step RL, significantly affecting learning. The prominence of off-policy bias intensifies in multi-step GCRL, especially as the reward accumulation steps increase, magnifying both bias and variance (Yang et al., 2021). Yang et al. (2021) presented MHER, MHER($\lambda$), and MMHER, which merge multi-step learning with hindsight relabeling (Andrychowicz et al., 2017). While vanilla MHER lacks mechanisms to handle off-policy bias, MHER($\lambda$) extends the MHER base by leveraging techniques from TD($\lambda$) (Sutton & Barto, 2018). MMHER, meanwhile, uses an on-policy target from a learned world model. The robustness of MHER($\lambda$) and MMHER hinges on MHER's inherent stability and the precision of the learned model, respectively. In contrast, our study introduces two novel subtypes of multi-step off-policy bias in GCRL: *shooting* and *shifting* biases. The latter, unique to GCRL, emerges from accumulating off-policy biases at goal states, influencing prior state value estimates. We found pronounced shifting biases in multi-step GCRL for intricate tasks. Our truncation technique effectively curtails this bias. Our insights suggest specific behavior policies might offer benefits. With this, we deploy quantile regression to address off-policy bias comprehensively.

**Quantile-Based Regression**    Quantile regression was pioneered by Koenker & Hallock (2001) and later adapted for distributional RL to model state-action value functions' quantile function (Dabney et al., 2018a;b; Kuznetsov et al., 2020). Expanding on this, Tang et al. (2022) integrated multi-step RL within distributional RL frameworks utilizing quantile regression. Motivated by these studies, our exploration specifically targets the upper quantiles of the value distribution. While rooted in deterministic settings, we account for stochasticity introduced by multi-step behavior policies, which sets our work apart from existing studies. Unlike prior works learning a complete stochastic value distribution, we prioritize the upper quantile, representing optimal outcomes under select effective behavior policies tied to goals. This quantile guides agents toward actions likely surpassing current estimates and counteracts suboptimal behavior policies' drawbacks.

## 3 PRELIMINARY

GCRL is framed as a goal-augmented MDP characterized by parameters $(\mathcal{S}, \mathcal{A}, \mathcal{T}, r, \mathcal{G}, p_{dg}, \phi, \gamma, T)$. Here, $\mathcal{S}$, $\mathcal{A}$, $\gamma$, and $T$ are state space, action space, discount factor, and horizon, respectively. The transition function $\mathcal{T}(s, a, s')$ gives the probability of transitioning from state $s$ to $s'$ via action $a$. The goal space is $\mathcal{G}$, with $p_{dg}$ as the desired goal distribution. The function $\phi(s)$ maps state $s$ to

its achieved goal. In sparse reward settings, the reward $r(s', g)$ is binary, being zero if the distance $d(\phi(s'), g)$ between achieved and target goals is less than $\epsilon$ and $-1$ otherwise. The success of an episode $\tau$ with goal $g$ is defined by $r(s_T^\tau, g) = 0$. GCRL's objective is to learn a goal-conditioned policy $\pi(s, g)$ maximizing:

$$J(\pi) = \mathbb{E}_{g \sim p_{dg}, a_t \sim \pi(s_t, g), s_{t+1} \sim \mathcal{T}(\cdot | s_t, a_t)} \left[ \sum_{t=0}^{T-1} \gamma^t r(s_{t+1}, g) \right].$$

Utilizing the actor-critic framework, where policy $\pi_\psi$ and action-value function $Q_\theta$ are parameterized by $\psi$ and $\theta$, respectively:

$$\pi_\psi(s, g) = \arg\max_a Q_\theta(s, a, g), \qquad Q_\theta(s, a, g) = \mathbb{E}_{t, \tau \sim \mathcal{B}} \left[ y^{(n)}(t, \tau, g) \Big| s_t^\tau = s, a_t^\tau = a \right] \tag{1}$$

where $y^{(n)}(t, \tau, g) = \sum_{i=1}^{n-1} \gamma^i r(s_{t+i}^\tau, g) + \gamma^n Q_{\bar{\theta}}(s_{t+n}^\tau, \pi_{\bar{\psi}}(s_{t+n}^\tau, g), g)$ is the $n$-step target value, and $s_t^\tau$ denotes the state at time $t$ of trajectory $\tau$. In this, $\bar{\psi}$ and $\bar{\theta}$ are the parameters of target actor and critic, as detailed in multi-step GCRL (Yang et al., 2021). Although $n$ can truncate if $t + n > T$, this is simplified for clarity. Typically, $\pi$ represents $\pi_\psi$, with emphasis on learning the action-value function. Further context is provided in Appendix A.

## 4  BIAS DECOMPOSITION

This section delves into the off-policy bias in multi-step GCRL. Through comprehensive analysis, we decompose this bias into two distinct types and examine their implications on action-value estimation.

The temporal difference (TD) error between two consecutive states at the step $t$ of trajectory $\tau$ is

$$\delta_\theta(s_t^\tau, a_t^\tau, g) = \gamma Q_\theta(s_{t+1}^\tau, \pi_\phi(s_{t+1}^\tau, g), g) + r(s_{t+1}^\tau, g) - Q_{\bar{\theta}}(s_t^\tau, a_t^\tau, g).$$

The $n$-step TD error between the $n$-step targets and the action-value function is characterized as

$$y^{(n)}(t, \tau, g) - Q_\theta(s_t^\tau, a_t^\tau, g) = \delta_\theta(s_t^\tau, a_t^\tau, g) + \sum_{i=1}^{n-1} \gamma^i [A_{\bar{\theta}}(s_{t+i}^\tau, a_{t+i}^\tau, g) + \delta_{\bar{\theta}}(s_{t+i}^\tau, a_{t+i}^\tau, g)], \tag{2}$$

where $A_{\bar{\theta}}(s_{t+i}^\tau, a_{t+i}^\tau, g) = Q_{\bar{\theta}}(s_{t+i}^\tau, a_{t+i}^\tau, g) - Q_{\bar{\theta}}(s_{t+i}^\tau, \pi(s_{t+i}^\tau, g), g)$ is the estimated advantage of action $a_{t+i}^\tau$ in state $s_{t+i}^\tau$, given goal $g$. The details of the derivation of Eq. (2) can be found in Appendix A. The learning target for $\delta_\theta(s, a, g)$ is formulated as

$$\delta_\theta(s, a, g) = \mathbb{E}_{t, \tau} \left[ \sum_{i=1}^{n-1} \gamma^i [A_{\bar{\theta}}(s_{t+i}^\tau, a_{t+i}^\tau, g) + \delta_{\bar{\theta}}(s_{t+i}^\tau, a_{t+i}^\tau, g)] \Big| s_t^\tau = s, a_t^\tau = a \right]. \tag{3}$$

Here, $\delta_\theta(s, a, g)$ encapsulates biases arising from both the off-policy advantage and TD error components. In the subsequent analysis, we will explore how this localized bias aggregates to form a broader off-policy bias in the action-value function.

For on-policy evaluation, let $\tau_\pi^g$ denote the trajectory resultant from policy $\pi(\cdot, g)$. Recursive decomposition of the learned action-value $Q_{\bar{\theta}}(s_t^{\tau_\pi^g}, a_t^{\tau_\pi^g}, g)$ and its unbiased counterpart $Q_\pi(s_t^{\tau_\pi^g}, a_t^{\tau_\pi^g}, g)$ along $\tau_\pi^g$ until episode termination yields:

$$Q_{\bar{\theta}}(s_t^{\tau_\pi^g}, a_t^{\tau_\pi^g}, g) = \sum_{i=t}^{T} \gamma^{i-t} [r_i^{\tau_\pi^g} + \delta_{\bar{\theta}}(s_i^{\tau_\pi^g}, a_i^{\tau_\pi^g}, g)] + \gamma^T Q_{\bar{\theta}}(s_T^{\tau_\pi^g}, \pi(s_T^{\tau_\pi^g}, g), g), \tag{4}$$

$$Q_\pi(s_t^{\tau_\pi^g}, a_t^{\tau_\pi^g}, g) = \sum_{i=t}^{T} \gamma^{i-t} r_i^{\tau_\pi^g} + \gamma^T Q_\pi(s_T^{\tau_\pi^g}, \pi(s_T^{\tau_\pi^g}, g), g). \tag{5}$$

In the context of multi-step GCRL, our aim is to devise a strategy that minimizes the detrimental effects of off-policy bias, thereby augmenting the agent's proficiency in achieving goals. Given this focus, off-policy bias in failing policies becomes less of interest given our target. Consequently, our

analysis intentionally focuses on successful evaluation trajectories, denoted as $\tau_\pi^{g+}$. Analyzing off-policy bias in successful trajectories, together with overall success rates, provides a multidimensional perspective. This approach not only gives a thorough assessment of the algorithm's performance but also reveals how bias adversely affects learning efficiency and success rates. For a successful trajectory $\tau_\pi^{g+}$, both $r_{T-1}^{\tau_\pi^{g+}}$ and $Q_\pi(s_T^{\tau_\pi^{g+}}, \pi(s_T^{\tau_\pi^{g+}}, g), g)$ are zero. Given these conditions and Eqs. (4) and (5), we discern the off-policy bias as:

$$B(s_t^{\tau_\pi^{g+}}, a_t^{\tau_\pi^{g+}}, g) = \underbrace{\sum_{i=t}^{T} \gamma^{i-t} \delta_{\bar\theta}(s_i^{\tau_\pi^{g+}}, a_i^{\tau_\pi^{g+}}, g)}_{\text{Shooting Bias}} + \underbrace{\gamma^T Q_{\bar\theta}(s_T^{\tau_\pi^{g+}}, \pi(s_T^{\tau_\pi^{g+}}, g), g)}_{\text{Shifting Bias}}. \tag{6}$$

Thus, we dissect the off-policy bias in Eq. (6) into two distinct segments: the *shooting bias* and the *shifting bias*. The shooting bias, accumulating within the finite time horizon $T$, can distort the action-value function and affect decision-making. The shifting bias manifests mainly within goal states which accumulate over paths of potentially infinite lengths, which we refer to as *bootstrap paths* in subsequent sections. Both biases are influenced by off-policy data through the same underlying mechanism, as illustrated in Eq. (3). While the shooting bias in multi-step GCRL shows similarities to off-policy biases in traditional multi-step RL, the shifting bias is unique to multi-step GCRL.

## 5 BIAS ANALYSIS AND MANAGEMENT

In this section, we analyze the shooting and shift biases, propose strategies to manage them, and introduce metrics to quantify them.

### 5.1 SHOOTING BIAS

To analyze the shoot bias, we first present the following proposition:

**Proposition 1.** Given a goal-conditioned action-value function $Q(\cdot, \cdot, \cdot)$, if the policy $\pi$ has been optimized based on $Q$, the associated advantage function $A(s_t^\tau, a_t^\tau, g) \leq 0$ for any $\tau$ and $t$.

**Justification:** In Eq. (1), if the policy $\pi_\psi$ is properly optimized according to the action-value function $Q_\theta$, we have

$$\pi_\psi(s, g) = \arg\max_a Q_\theta(s, a, g) = \arg\max_a A_\theta(s, a, g), \qquad A_\theta(s, \pi_\psi(s, g), g) = 0.$$

Therefore, the advantages along on-policy trajectories should be zeros. Based on Eq. (2), the $n$-step TD errors between the target value calculated from the on-policy trajectories $\tau_\pi^{g+}$ can be simplified as

$$y^{(n)}(t, \tau_\pi^{g+}, g) - Q_\theta(s_t^{\tau_\pi^{g+}}, a_t^{\tau_\pi^{g+}}, g) = \delta_\theta(s_t^{\tau_\pi^{g+}}, a_t^{\tau_\pi^{g+}}, g) + \sum_{i=1}^{n-1} \gamma^i \delta_{\bar\theta}(s_{t+i}^{\tau_\pi^{g+}}, a_{t+i}^{\tau_\pi^{g+}}, g).$$

The on-policy $n$-step TD error is unbiased while the acquired $\delta_\theta(s_t^{\tau_\pi^{g+}}, a_t^{\bar\tau_\pi^{g+}}, g)$ in Eq. (3) is biased by the off-policy data. In practice, this bias often tends negative due to the presence of suboptimal behavior policies. This results in a systematic underestimation of each action's expected return.

A direct approach to mitigate the bias issue is the implementation of importance sampling (IS) methods (Precup et al., 2001), aiming to correct all sources of shooting bias. Leaving out the variance problem it may bring, IS may not fully harness multi-step targets to enhance learning efficiency. For deriving unbiased targets, the agent might significantly truncate the step count, particularly evident in policies exhibiting low entropy. In extreme scenarios, multi-step learning for a deterministic policy could revert to one-step learning. Though it appears reasonable to disdain the off-policy data less irrelevant to the current policy, we argue that it is worth rethinking a pivotal question: *Is bias always detrimental, or can specific biases positively guide and improve the target policy?*

While off-policy bias can disrupt the learning process, the ultimate goal of RL is maximizing the expected return. This context prompts a critical inquiry: *Can off-policy biases from behavior policies, especially those indicating optimal multi-step actions, positively influence learning efficiency and enhance the policy?*

In the off-policy $n$-step TD error as shown in Eq. (2), $\sum_{i=1}^{n-1} \gamma^i [A_{\bar{\theta}}(s_{t+i}^\tau, a_{t+i}^\tau, g) + \delta_{\bar{\theta}}(s_{t+i}^\tau, a_{t+i}^\tau, g)]$ can be seen as the rectified advantage terms by considering the TD errors. For an on-policy state-action pair $(s_t^{\tau_\pi^{g+}}, a_t^{\tau_\pi^{g+}})$, assume existence of a $t'$ and $\tau'$ where $y^{(n)}(t', \tau', g) > y^{(n)}(t, \tau_\pi^{g+}, g)$, $s_{t'}^\tau = s_t^{\tau_\pi^{g+}}$ and $a_{t'}^\tau = a_t^{\tau_\pi^{g+}}$. It can be inferred that

$$\sum_{i=1}^{n-1} \gamma^i [A_{\bar{\theta}}(s_{t'+i}^\tau, a_{t'+i}^\tau, g) + \delta_{\bar{\theta}}(s_{t'+i}^\tau, a_{t'+i}^\tau, g)] \geq \sum_{i=1}^{n-1} \gamma^i \delta_{\bar{\theta}}(s_{t+i}^{\tau_\pi^{g+}}, a_{t+i}^{\tau_\pi^{g+}}, g).$$

In this scenario, the behavior policy's actions over the next $n$ steps yield enhanced rectified advantages relative to the current policy, emphasizing their greater goal-reaching efficiency. We can transition to learning the value functions of this superior policy, a combination of the behavior policy over the $n$ steps followed by the target policy. We identify this off-policy bias as beneficial, setting it apart from harmful biases where target values are less than on-policy targets. Even if no such $\tau'$ and $t'$ exist, implying all off-policy biases are detrimental, data with higher target values still result in less harmful bias. Thus, maximizing target values leads to either more beneficial or reduced off-policy bias than computing the expectation; i.e.,

$$Q_\theta^{max}(s, a, g) = \max_{t, \tau} \left\{ y^{(n)}(t, \tau, g) \mid s_t^\tau = s, a_t^\tau = a \right\}.$$

However, obtaining $Q_\theta^{max}(s, a, g)$ is complex in deep RL when using neural networks to approximate the action-value function. In both high-dimensional and continuous tasks, identifying all off-policy data and their maximum is challenging. By setting the target values as the maximum between the current estimation and the target value, an over-optimistic bias can emerge. This might hinder the action-value function's adaptability to changes in target value distributions. For example, action-values typically start near zero after neural network initialization and can decrease significantly to be negative during learning. Such an approach might not capture the true maximum value accurately.

**Bias Resilience with Quatile Regression**    Rather than pursuing the maximum, we reorient our focus towards learning the action-value function to capture an upper quantile $\rho$ ($0.5 < \rho < 1$) of the target values. This approach maintains the advantage over taking the expectation of target values by emphasizing beneficial off-policy bias and diminishing detrimental ones. Additionally, it ensures adaptability to shifts in the target value distribution, ensuring consistent responsiveness to action-value dynamics. To learn the upper quantile, we employ quantile regression (Koenker & Hallock, 2001; Dabney et al., 2018b). Although parameters are essential, they are omitted here for clarity. The combined loss, integrating both the quantile level $\rho$ and the Huber loss (Huber, 1992) threshold $\kappa$, is defined as:

$$\mathcal{L}_{QR}(y, Q_\theta^\rho; \rho, \kappa) = |\rho - \mathbb{I}(y, Q_\theta^\rho)| \cdot \begin{cases} (y - Q_\theta^\rho)^2 & \text{if } |y - Q_\theta^\rho| \leq \kappa, \\ \kappa \left(2 \cdot |y - Q_\theta^\rho| - \kappa\right) & \text{else.} \end{cases} \tag{7}$$

In Eq. (7), $Q_\theta^\rho$ represents the action-value function obtained through quantile regression. The indicator function $\mathbb{I}(y, Q_\theta^\rho) = 1$ if $y < Q_\theta^\rho$, and 0 otherwise. To further minimize the off-policy bias problems in multi-step GCRL, we incorporate MHER($\lambda$) (Yang et al., 2021) to derive target values by interpolating multi-step targets across various time steps.

## 5.2    Shifting Bias

Shifting bias, emerging as a special form of shooting bias, accumulates in goal states over an infinite horizon. To facilitate the analysis of shifting bias, we define a bootstrap path of infinite horizon.

**Definition 1** (Bootstrap Path). Let $\beta_{(s,a)}$ represent a *bootstrap path*, a trajectory clarifying how rewards are retroactively sent to the current state. In the context of $n$-step multi-step learning, the bootstrap path for tuple $(s, a, g)$ is recursively formed as outlined below: **1) Initialization:** Initiate by establishing the bootstrap path, represented as $\beta_{(s,a)}$, by employing an $n$-step transition $(s_t^\tau, a_t^\tau, \cdots, s_{t+n}^\tau)$ wherein $(s_t^\tau, a_t^\tau) = (s, a)$. Designate the anchor state-action pair as $(\hat{s}, \hat{a}) = (s_{t+n}^\tau, \pi(s_{t+n}^\tau, g))$. **2) Selection:** Randomly select another $n$-step transition $(s_{t'}^{\tau'}, a_{t'}^{\tau'}, \cdots, s_{t'+n}^{\tau'})$ such that $(s_{t'}^{\tau'}, a_{t'}^{\tau'}) = (\hat{s}, \hat{a})$. **3) Merge:** Extend $\beta_{(s,a)}$ by appending the $n$-step transition to its end. Thereafter, reset the anchor state-action pair as $(\hat{s}, \hat{a}) = (s_{t'+n}^{\tau'}, \pi(s_{t'+n}^{\tau'}, g))$. **4) Iteration:** Persistently execute steps 2) and 3).

Even without full knowledge of $\beta_{(s,a)}$, we know the learning objective of the action-value function in $s_g$ is non-positive, as shown in Eq. (8). This is due to non-positive sparse rewards in the environment:

$$Q_\theta(s_g, a, g) = \mathbb{E}_{\beta_{(s_g,a)}} \left[ \sum_{i=1}^\infty \gamma^i r(s_i^{\beta_{(s_g,a)}}, g) \right] \le 0.$$

Here, $s_i^{\beta_{(s_g,a)}}$ is the $i$th state in $\beta_{(s_g,a)}$. Regardless of the optimality of action $a$, negative rewards arise if any action in a bootstrap path diverts the agent from the goal. This is common, especially with hindsight relabelling techniques (Andrychowicz et al., 2017). With goal relabelling, the goal state may serve as a waypoint to another goal. Actions that deviate the agent can intensify the bias shift. The iterative learning towards the objective magnifies this bias, altering learning targets of preceding state-action pairs and causing unstable learning.

**Truncated Multi-Step Targets**   To mitigate the shifting bias, we propose truncating the multi-step transitions at the first goal state, thereby excluding actions of the behavior policy that lead the agent away from the goal. The truncated multi-step target denoted by $\hat{y}^{(n)}$ is calculated as follows:

$$\hat{y}_\pi^{(n)}(t, \tau, g) = r(s_{t+1}^\tau, g) + \begin{cases} \gamma \hat{y}_\pi^{(n-1)}(t+1, \tau, g) & \text{if } r(s_{t+1}^\tau, g) \ne 0 \text{ and } n > 1, \\ \gamma Q_\pi^{(n)}(s_{t+1}^\tau, \pi(s_{t+1}^\tau, g), g) & \text{otherwise.} \end{cases}$$

Since the agent is already in the goal states, the need for rapid reward propagation through multi-step learning is naturally reduced. This method emphasizes the importance of minimizing shifting bias to ensure stable learning targets for preceding state-action pairs.

## 5.3 BIAS MEASUREMENT

In analyzing off-policy bias, we aim to quantify shooting and shifting biases as in Eq. (6), introducing metrics *Terminal Shifting Bias* (TSB) and *Initial Shooting Bias* (ISB) defined as follows:

$$TSB_\pi = \mathbb{E}_{g, \tau_\pi^{g+}} \left[ Q_\pi(s_0^{\tau_\pi^{g+}}, a_0^{\tau_\pi^{g+}}, g) \right], \quad ISB_\pi = \mathbb{E}_{\tau_\pi^{g+}, g} \left[ Q_\pi(s_0^{\tau_\pi^{g+}}, a_0^{\tau_\pi^{g+}}, g) \right] - \gamma^T \cdot TSB_\pi.$$

In the metrics, TSB reflects the severity of the shifting bias by measuring the discrepancy between 0 and the action-value prediction of the terminal state in successful evaluation trajectories. Furthermore, ISB captures the shooting bias in the initial state, accumulated over the entire episode horizon.

In summary, this section answers two research questions via an in-depth analysis and presents novel strategies for mitigating off-policy biases. These insights and approaches are consolidated into the *Bias-Resilient MHER* (BR-MHER) algorithm, detailed in Appendix B.

## 6 EXPERIMENTS

In this section, we evaluate the performance of our BR-MHEM algorithm. The evaluation metrics for the algorithm include sampling efficiency, success rates and off-policy biases. The performance of our approaches is further exemplified through a comparative investigation. In our experiments, we use six robotic tasks involving three distinct robotic agents: i) Fetch robotic arm (Plappert et al., 2018): `FetchPickAndPlace` and `FetchSlide`; ii) Anthropomorphic robotic hand (Plappert et al., 2018): `HandReach` and `HandManipulateBlockRotateXYZ`; iii) Sawyer robotic arm (Nair et al., 2018): `SawyerPushAndReachEnvEasy` and `SawyerReachXYZEnv`. We also apply our methods in the grid-world tasks `Simple-MiniGrid-Empty` (Chevalier-Boisvert et al., 2018) of scales 25×25 and 50×50. For more details about these tasks, see Appendix C.

## 6.1 BASELINES

For a comparative study, we include six experimental baselines: i) HER (Andrychowicz et al., 2017): A straightforward GCRL technique using one-step targets; ii) MHER (Yang et al., 2021): An expanded version of HER learning through multi-step targets; iii) MHER($\lambda$) (Yang et al., 2021): A balanced method similar to TD($\lambda$) (Sutton & Barto, 2018), harmonizing multi-step targets at various steps via the $\lambda$ parameter; iv) MMHER (Yang et al., 2021): A model-based approach, calculating on-policy multi-step targets with a world model; v) IS-MHER: A integration of MHER and importance sampling method Retrace($\lambda$) (Munos et al., 2016); vi) WGCSL (Yang et al., 2022): An advanced goal-conditioned supervised learning algorithm with meticulously designed weighting mechanisms.

## 6.2 EXPERIMENTAL SETTINGS AND IMPLEMENTATION

**Experimental Settings**   Our experiments aim to answer the following research questions: **Q1)** How effectively does our BR-MHER manage off-policy bias in multi-step GCRL? **Q2)** How do our bias mitigation techniques impact learning efficiency? **Q3)** Does BR-MHER maintain competitive performance with state-of-the-art methods such as IS-MHER and WGCSL in terms of success rate and other relevant metrics? Additionally, an ablation study is conducted to provide deeper insights into our proposed techniques, see Appendix D.

We conduct comprehensive experiments, benchmarking our BR-MHER against the baselines across a variety of robotic tasks with $n = 3, 5, 7$, and grid-world tasks with $n = 3, 5, 10$. To ensure the reliability of our results, we perform five trials with distinct seeds for each task. Training is stopped after 50 epochs, and the performance is assessed based on the success rate along with the bias and vairance metrics, as elaborated in Section 5.3.

**Implementation**   We utilize a modified OpenAI Baselines codebase (Dhariwal et al., 2017), following the recommendations made by Yang et al. (Yang et al., 2021). To mitigate over-estimation bias, we adopt the Clipped Double Q-learning (CDQ) and delayed policy updates from TD3 (Fujimoto et al., 2018) for all GCRL methods. Our implementation utilizes Multilayer Perceptrons (MLPs) for both actor and critic networks. For a balanced comparison with quantile regression, we used Huber loss for all GCRL methods. The upper quantile ($\rho$) is set at 0.75, and the Huber loss threshold ($\kappa$) is 10. The upper quantile ($\rho$) is fixed at 0.75 to balance capturing the upper quantile and adjusting target value distribution shifts. The Huber loss threshold ($\kappa$) is set at 10. Detailed specifications of these hyper-parameters and distinct network configurations for various tasks are in Appendix C.

## 6.3 EXPERIMENT RESULTS

Due to the limited space, we report only the statistical results (mean and standard deviation) across five random seeds for the multi-step GCRL methods with the largest $n$, specifically 7 for robotic tasks and 10 for grid-world tasks. Additional results and an ablation study are in Appendix D.

**Comparison with HER**   As illustrated in Fig.1(a), BR-MHER achieves higher success rates than HER in four tasks and equivalent performance in the remaining tasks. Additionally, BR-MHER outperforms HER in learning efficiency on six tasks and matches efficiency in both `FetchSlide-v1` and `HandManipulateBlockRotateXYZ-v0` tasks. Our method attains a shooting bias comparable to HER, as denoted by ISB in Fig.1(b). Regarding shifting bias, highlighted by TSB in Fig. 1(c), BR-MHER records smaller values in six tasks and analogous values in the other two, signifying that even unbiased single-step learning may accumulate errors, detectable by TSB, over an extensive horizon. It also demonstrates that our method effectively mitigates such intrinsic bias in GCRL, alongside off-policy shifting bias.

**Comparison with MHER**   MHER, a baseline multi-step GCRL method, exhibits high sensitivity to off-policy bias. Fig.1(a) demonstrates the superior performance of BR-MHER in success rates and learning efficiency across six tasks. However, in cases where both methods achieve similar success rates, MHER holds a slight edge in learning efficiency, exclusively in the Sawyer tasks. A detailed bias analysis further highlights consistently larger off-policy bias magnitudes in all tasks by MHER, corroborated by the ISB and TSB metrics in Fig.1(b) and 1(c). This bias issue, however, is least severe in the Sawyer tasks, allowing MHER to maintain decent performance.

**Comparison with MMHER**   MMHER is model-based, in contrast to the model-free approach of BR-MHER. Fig.1(a) shows that BR-MHER generally outperforms or matches MMHER across various tasks, except in `FetchSlide-v1`. This task's unique nature, where actions have limited impact once an object is slid, allows future states to predominantly dictate the return. These types of transitions can be swiftly learned by the world model, granting MMHER a notable learning advantage. Despite this, BR-MHER demonstrates reduced magnitudes of TSB and ISB across all tasks, indicative of lower off-policy biases as seen in Fig.1(b) and 1(c). While MMHER employs on-policy rollouts to avoid these biases, it contends with accumulating model-prediction errors, evidenced in the Fetch tasks in Fig. 1(b), where an initial increase in ISB possibly reflects heightened model-prediction errors before decreasing as the model better adapts to the environments.

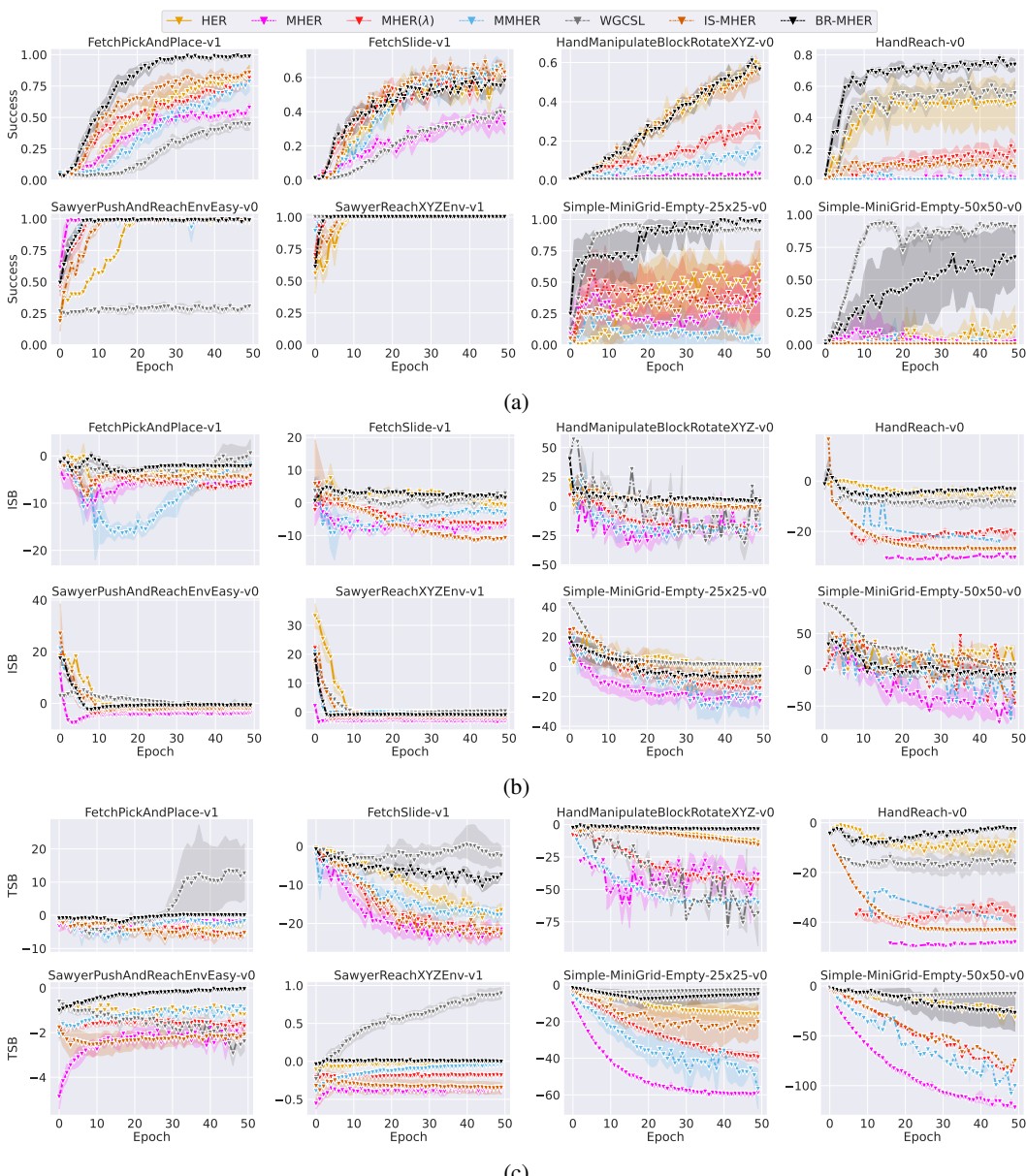

Figure 1: Comparative study of seven-step ($n$=7) GCRL methods on robotic tasks and ten-step ($n$=10) GCRL methods on grid-world tasks. (a) Success rate. (b) ISB for measuring shooting bias. (c) TSB for measuring shifting bias.

**Comparison with MHER($\lambda$)** BR-MHER achieves superior success rates and learning efficiency across five diverse tasks and maintains comparable performance on the Sawyer tasks, as demonstrated in Fig.1(a). Despite a slight disadvantage in the `FetchSlide-v1` task, the ablation study in Appendix D reveals that the degradation occurs with the introduction of quantile regression. Despite its significant mitigation of TSB bias, it poses a challenge in this specific task. In `FetchSlide-v1`, a state within the target object's sliding process directly determines future rewards since actions can no longer influence the target object, leading to a largely deterministic distribution of target values. This scenario causes the quantile regression to prefer positive network approximation errors in the target value, culminating in a slightly optimistic shooting bias, as evident by the ISB shown in Fig.1(b), and likely hindering learning. Despite this, Figs.1(b) and 1(c) confirm that BR-MHER significantly lowers both ISB and TSB, highlighting its ability to mitigate off-policy biases.

**Comparison with IS-MHER**   BR-MHER outperforms IS-MHER in learning efficiency and success rates across six tasks, showing similar performance in `HandManipulateBlockRotateXYZ-v0`, but slightly inferior in `FetchSlide-v1`, as shown in Fig.1(a). IS-MHER exhibits larger TSB and ISB magnitudes, outlined in Figs.1(b) and 1(c). This trend is partly due to the evaluation of a deterministic policy, derived from the stochastic policy used in importance sampling, focused on optimizing success rate evaluations. The disparity is further exacerbated by the use of truncated importance ratios in Retrace($\lambda$) (Munos et al., 2016).

**Comparison with WGCSL**   Our comparative assessment highlights a notable inconsistency in WGCSL's performance across various tasks. Despite exhibiting enhanced learning efficiency in grid-world tasks, as depicted in Fig.1(a), it falls behind BR-MHER in success rates for five robotic tasks. As observed in Fig. 1(c), BR-MHER holds smaller magnitudes of ISB, indicating less severe shooting bias compared to IS-MHER. Although WGCSL employs an action-value function learned via single-step learning, it still exhibits significant shifting biases in certain tasks. This is evidenced by the elevated TSB in `FetchPickAndPlace-v1` and `HandManipulateBlockRotateXYZ-v0` (Fig.1(c)), correlating with its subpar performance. This discrepancy stems from the supervised policy's insufficient synchronization with the action-value function's directives, inducing substantial policy and target value fluctuations during the learning process. In contrast, our GCRL methods adeptly mitigate this issue, demonstrating reduced TSB in robotic tasks. In grid-world tasks, WGCSL displays markedly lower TSB, indicating diminished fluctuations. In `Simple-MiniGrid-Empty-50x50`, BR-MHER encounters notable TSB variation among different seeds, leading to a stark success rate oscillation between over 95% and approximately 30%. This variation might be attributed to the imbalance between learning action-values in goal states with low variance of target values and states preceding the goals with higher target value variance. This imbalance occasionally prevents the effective minimization of TD errors in goal states, leading to a high magnitude of shifting bias. This observed variation in success rates and the shifting bias underscores BR-MHER's limitations in handling shifting bias effectively in this specific scenario.

In response to the proposed research questions: **A1)** BR-MHER substantially minimizes both shooting and shifting bias compared to MHER($\lambda$) and other GCRL methods, as evidenced by the ISB and TSB metrics. **A2)** BR-MHER generally excels over MHER($\lambda$), achieving enhanced success rates and learning efficiency across multiple tasks, albeit with a slight setback in the `FetchSlide-v1` task, potentially attributed to over-optimistic shooting bias induced by quantile regression. **A3)** While IS-MHER and WGCSL exhibit superior performance in specific tasks (`FetchSlide-v1` and two grid-world tasks respectively), BR-MHER consistently delivers solid and reliable performance across all other tasks and experiments, highlighting its robust and adaptable nature in various contexts.

## 7   CONCLUSION

This paper provides a robust exploration into the off-policy bias challenges inherent in multi-step GCRL. By uncovering and thoroughly analyzing two types of off-policy biases present in multi-step GCRL, we have shed light on innovative techniques for regulating these biases. This led to the inception of our resilient multi-step GCRL algorithm, BR-MHER. Combined metrics, encapsulating aspects of performance and off-policy bias, robustly showcases the resilience and robustness of our BR-MHER against bias issues, especially as the $n$ increases in the $n$-step targets. Notably, our experimental evaluations robustly demonstrates the superiority of BR-MHER over the state-of-the-art MHER($\lambda$) and other advanced multi-step baselines for a majority of the GCRL benchmarks in most tasks, both in terms of success rate and learning efficiency. Although our analysis primarily pertains to a deterministic environment, it underscores the potential of harnessing beneficial off-policy bias instead of avoiding all off-policy bias altogether.

The presented research underpins the critical need for discerning the nature of biases in multi-step GCRL and postulates that not all biases are detrimental. Recognizing and leveraging these biases can be transformative, carving a path for superior, adaptive, and efficient algorithms. It redefines the approach towards bias in multi-step GCRL, potentially unlocking expansive applications across a myriad of complex environments. Moreover, while our approach demonstrates robust performance across a variety of tasks, our observations in certain specific scenarios highlight opportunities for further refinement and enhancement of our strategies. Our ongoing work aims to explore a more robust strategy for off-policy bias mitigation and extend our approach to stochastic environments.

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
