# SUPPLEMENTARY MATERIALS FOR
# BIAS RESILIENT MULTI-STEP OFF-POLICY
# GOAL-CONDITIONED REINFORCEMENT LEARNING

## Appendix A:    Background

This appendix offers an in-depth exploration of the technical aspects of TD3, HER, MHER, MHER($\lambda$), MMHER, Retrace($\lambda$), and WGCSL. Additionally, we furnish a detailed derivation of the learning objective of TD error in multi-step GCRL, as delineated in Eq. (3) from the main manuscript.

### A.1    TWIN DELAYED DEEP DETERMINISTIC POLICY GRADIENT (TD3)

TD3 (Fujimoto et al., 2018) is an off-policy reinforcement learning method with an actor-critic architecture designed for continuous control tasks. The actor $\pi_\psi$ is parameterized by $\psi$, and there are two critics $Q_{\theta_1}$ and $Q_{\theta_2}$ parameterized by $\theta_1$ and $\theta_2$, respectively. Given a data tuple $(s, a, s', r)$, the target value is calculated as

$$y(s, a, s', r) = r + \gamma \min_{i=1,2} Q_{\bar{\theta}_i}(s', \pi_{\bar{\psi}}(s'))$$

This approach is also known as the Clipped Double Q-learning (CDQ) algorithm used to mitigate over-optimistic bias in the learning process. Unlike traditional actor-critic algorithms that update the actor and critics simultaneously, TD3 proposes to delay the actor updates until the action-value errors are small. This delay is typically implemented by setting the actor update period $k$ to be larger than one. The training loss for the critic is represented by the expected $L2$ loss between the $Q$-value predictions and the target values $y$ as

$$L_{\theta_i} = \mathbb{E}_{(s,a,s',r)\sim\mathcal{B}}[(y(s, a, s', r) - Q_{\theta_i}(s', a))^2], \ i \in \{1, 2\}$$

Every $k$ training steps, the actor is optimized by maximizing the $Q$-value of the action output $\pi_\psi(s)$ for the given states, and the corresponding loss is

$$L_\psi = -\mathbb{E}_{s\sim\mathcal{B}}[Q_{\theta_1}(s, \pi_\psi(s))]$$

In the following discussion of algorithms, we omit parameters when they are not necessary, prioritizing clarity and simplicity.

### A.2    HINDSIGHT EXPERIENCE REPLAY (HER)

HER (Andrychowicz et al., 2017) proposes to tackle sparse learning signals in sparse-reward settings via relabeling the goal as the actually achieved goals in the future. Consider a goal-conditioned data tuple $(s_t^\tau, a_t^\tau, s_{t+1}^\tau, g^\tau)$, extracted from the trajectory $\tau$. The goals actually achieved in the future constitute the set $\{\phi(s_{t'}^\tau)\}_{t'=t+1}^T$. Under HER, the original goal $g^\tau$ is relabeled as $\hat{g}^\tau$, selected from the distribution $p(\{\phi(s_{t'}^\tau)\}_{t'=t+1}^T)$. This distribution is dependent on the hindsight relabeling strategy and is typically a uniform distribution over $\{\phi(s_{t'}^\tau)\}_{t'=t+1}^T$.

Consequently, the agent learns from the relabeled data tuple $(s_t^\tau, a_t^\tau, s_{t+1}^\tau, \hat{g}^\tau)$ with a recalculated reward $r(s_{t+1}^\tau, \hat{g}^\tau)$. This process ensures that the agent, at some point, receives valuable rewards propagated from the relabeled goal $\hat{g}^\tau$. The target value based on the relabeled goal is computed as:

$$y(s_t^\tau, a_t^\tau, s_{t+1}^\tau, \hat{g}^\tau) = r(s_{t+1}^\tau, \hat{g}^\tau) + \gamma Q_\pi(s_{t+1}^\tau, \pi(s_{t+1}^\tau, \hat{g}^\tau), \hat{g}^\tau), \tag{A1}$$

where the subscript $\pi$ of $Q_\pi$ denotes that the target values for the action-value function are bootstrapped with the action decision from policy $\pi$. With the hindsight relabeling strategy for trajectory

$\tau$ at time $t$ represented by a stochastic function $h(t,\tau)$, where the relabeled goal $\hat{g}^\tau$ is sampled from the distribution defined by $h(t,\tau)$, the loss for the goal-conditioned action-value function can be expressed as:

$$L_{Q_\pi} = \mathbb{E}_{(s_t^\tau, a_t^\tau, s_{t+1}^\tau, g_t^\tau) \sim \mathcal{B}, \hat{g}^\tau \sim h(t,\tau)}[(y(s_t^\tau, a_t^\tau, s_{t+1}^\tau, \hat{g}^\tau) - Q_\pi(s_t^\tau, a_t^\tau, \hat{g}^\tau))^2]. \qquad \text{(A2)}$$

Then the goal-conditioned policy learns to optimize the expected $Q$-value function of $\pi(s_t^\tau, \hat{g}^\tau)$ for given states $s_t^\tau$ and corresponding goals $\hat{g}^\tau$ as

$$L_\pi = \mathbb{E}_{(s_t^\tau, g_t^\tau) \sim \mathcal{B}, \hat{g}^\tau \sim h(t,\tau)}[-Q_\pi(s_t^\tau, \pi(s_t^\tau, \hat{g}^\tau), \hat{g}_t^\tau)]. \qquad \text{(A3)}$$

### A.3 Multi-Step Hindsight Experience Replay (MHER)

In comparison to HER, MHER (Yang et al., 2021) integrates HER into multi-step GCRL with multi-step relabeling. Unlike Eq. (A1) that calculates the target value on a single-step transition, MHER calculates the target values on $n$-step transitions with the goal $g$ as follows

$$y^{(n)}(t,\tau,g) = \sum_{i=0}^{n-1} \gamma^i r(s_{t+i+1}^\tau, g) + \gamma^n Q_\pi(s_{t+n}^\tau, \pi(s_{t+n}^\tau, g), g). \qquad \text{(A4)}$$

MHER optimizes the actor and critic in similar ways as Eq. (A2) and Eq. (A3) except the target values are calculated via Eq. (A4). MHER enables the agent to learn faster from $n$-step targets.

### A.4 MHER($\lambda$)

Inspired by TD($\lambda$) (Sutton & Barto, 2018), MHER($\lambda$) (Yang et al., 2021) provides a novel approach to balance the trade-off between fewer-step targets that come with lower off-policy bias and multi-step targets, which carry more reward information. This balancing act is achieved through the introduction of an exponential decay weight parameter $\lambda$, where $\lambda \in [0,1]$. The multi-step targets, balanced by the $\lambda$ parameter, are calculated using the following formula:

$$y_\lambda^{(n)}(t,\tau,g) = \frac{\sum_{i=1}^n \lambda^i y^{(i)}(t,\tau,g)}{\sum_{i=1}^n \lambda^i}. \qquad \text{(A5)}$$

In this formulation, $y_\lambda^{(n)}$ leans towards the one-step target as $\lambda \to 0$, while higher weights are assigned to the $n$-step target as $\lambda$ increases.

### A.5 Model-Based MHER (MMHER)

MMHER (Yang et al., 2021) calculates on-policy multi-step targets from multi-step transitions generated via a world-model $M$ of the environment. The world-model $M$ is trained by past experiences. Given a one-step transition $(s_t^\tau, a_t^\tau, s_{t+1}^\tau)$, the agent simulates $(n-1)$-step on-policy transitions starting from $s_{t+1}^\tau$, following policy $\pi$, in order to pursue the goal $g$ through the world-model $M$. Within these $(n-1)$-step on-policy transitions, the $i$th $(i = 1 \cdots n-1)$ state generated from $M$ is denoted as $M_i^{\pi_g}(s_{t+1}^\tau)$, corresponding to the original $s_{t+1+i}^\tau$ in the off-policy data. Then, the multi-step target in MMHER is expressed as

$$y_M^{(n)}(s_t^\tau, a_t^\tau, s_{t+1}^\tau, g) = r(s_{t+1}^\tau, g) + \sum_{i=1}^{n-1} \gamma^i r(M_i^{\pi_g}(s_{t+1}^\tau), g)$$
$$+ \gamma^n Q_\pi(M_{n-1}^{\pi_g}(s_{t+1}^\tau), \pi(M_{n-1}^{\pi_g}(s_{t+1}^\tau), g), g). \qquad \text{(A6)}$$

However, $y_M^{(n)}$ actually suffers from model bias caused by the prediction errors of $M$, which can be severe when the problems are complex. To balance the model bias and learning information, Yang et al. (2021) proposes to integrate $n$-step model-based target and one-step target via a hyper-parameter $\alpha$ as

$$\bar{y}_M^{(n)}(s_t^\tau, a_t^\tau, s_{t+1}^\tau, g) = \frac{\alpha y_M^{(n)}(s_t^\tau, a_t^\tau, s_{t+1}^\tau, g) + r(s_{t+1}^\tau, g) + \gamma Q_\pi(s_{t+1}^\tau, \pi(s_{t+1}^\tau, g), g)}{1 + \alpha}. \qquad \text{(A7)}$$

The $\bar{y}_M^{(n)}(s_t^\tau, a_t^\tau, s_{t+1}^\tau, g)$ will serve as the final $n$-step targets for the action-value function to learn.

A.6    WEIGHTED GOAL-CONDITIONED SUPERVISED LEARNING (WGCSL)

The Weighted Goal-Conditioned Supervised Learning (WGCSL) algorithm, as delineated by Yang et al. (2022), stands as a supervised learning methodology designed for the execution of goal-conditioned tasks. This innovative algorithm employs goal relabeling techniques analogous to those used in Hindsight Experience Replay (HER) (Andrychowicz et al., 2017).

The formal supervised learning objective, $L_\psi^{WGCSL}$, is expressed as:

$$L_\psi^{WGCSL} = \mathbb{E}_{(s_t^\tau, a_t^\tau, g_t^\tau) \sim \mathcal{B}, \phi(s_i^\tau) \sim h(t, \tau)}[w_{t,i}^\tau \log \pi(a_t^\tau | s_t^\tau, \phi(s_i^\tau))].$$

In this equation, $s_i^\tau$ denotes the prospective state of $s_t^\tau$ where $i > t$, with the relabeled goal represented as $\phi(s_i^\tau)$, and the weight $w_{t,i}^\tau$ accentuating the prioritization of actions possessing a larger potential.

The weight $w_{t,i}^\tau$ is further defined as:

$$w_{t,i}^\tau = \gamma^{i-t} \cdot \text{clip}(\exp(A(s_t^\tau, a_t^\tau, \phi(s_i^\tau))), 0, K) \cdot \epsilon(A(s_t^\tau, a_t^\tau, \phi(s_i^\tau))). \tag{A8}$$

Here, $\gamma^{i-t}$, $\text{clip}(\exp(A(s_t^\tau, a_t^\tau, \phi(s_i^\tau))), 0, K)$, and $\epsilon(A(s_t^\tau, a_t^\tau, \phi(s_i^\tau)))$ symbolize the discounted relabeling weight (DRW), goal-conditioned exponential advantage weight (GEAW), and best-advantage weight (BAW) respectively. The DRW prioritizes learning based on the actions of states temporally closer to the relabeled goal. The GEAW emphasizes learning on actions with a higher advantage, employing $K$ to limit values in extreme cases. The BAW addresses the challenge of multi-modality in goal-conditioned Reinforcement Learning (RL), guiding the policy towards the modal with the maximal return, circumventing convergence to a weighted average of multiple modals. BAW is specifically formulated as:

$$\epsilon(A(s_t^\tau, a_t^\tau, \phi(s_i^\tau))) = \begin{cases} 1, & \text{if } A(s_t^\tau, a_t^\tau, \phi(s_i^\tau)) > \hat{A} \\ \epsilon_{\min}, & \text{otherwise} \end{cases}$$

Within this context, $\hat{A}$ denotes a threshold calculated from recent advantage statistics, while $\epsilon_{\min}$ represents a minimal positive value.

A.7    RETRACE($\lambda$)

Retrace($\lambda$) is a reinforcement learning algorithm specifically designed for off-policy correction of the value estimates (Munos et al., 2016). The algorithm employs importance sampling (IS) for correction, incorporating a safety mechanism to ensure the stability and robustness of learning. This mechanism involves truncating the importance sampling ratios to mitigate excessive variance in the estimates.

The $n$-step goal-conditioned target value based on Retrace($\lambda$) is calculated as

$$y_{\text{ret}}^{(n)}(t, \tau, g) = \sum_{i=1}^n \gamma^{i-1} \left( \prod_{s=1}^{i-1} c_s \right) \left( r(s_{t+i}^\tau, g) + \gamma \mathbb{E}_\pi[Q_\pi(s_{t+i}^\tau, \cdot)] - \gamma c_i Q(s_{t+i}^\tau, a_{t+i}^\tau) \right),$$

where $c_i$ is the truncated importance sampling ratio. It is defined that $\prod_{s=1}^0 c_s = 1$ and $c_n = 0$ as the experience beyond the $n$ steps is truncated. Specifically, the importance sampling ratio $c_t$ is:

$$c_i = \lambda \min \left( 1, \frac{\pi(a_{t+i}^\tau | s_{t+i}^\tau)}{\mu(a_{t+i}^\tau | s_{t+i}^\tau)} \right), \tag{A9}$$

where $\mu(a_t | s_t)$ is the behavior policy that generated the action $a_t$ at state $s_t$. The truncated importance sampling enables the agent to safely use off-policy data to correct the estimates.

A.8    TD ERROR LEARNING OBJECTIVE IN MULTI-STEP GCRL

This section meticulously delineates the derivation of the objective, depicted as Eq. (3) in the main text.

$$
\begin{aligned}
&Q_\theta(s_{t+1}^\tau, a_{t+1}^\tau, g) \\
&= r_t + \gamma Q_{\bar\theta}(s_{t+1}^\tau, \pi_{\bar\psi}(s_{t+1}^\tau, g), g) - \delta_\theta(s, a, g) \\
&= r_t + \gamma Q_{\bar\theta}(s_{t+1}^\tau, a_{t+1}^\tau, g) - \gamma A_{\bar\theta}(s_{t+1}^\tau, a_{t+1}^\tau, g) - \delta_\theta(s, a, g) && \text{(A10)} \\
&= r_t + \gamma \left[ r_{t+1} + \gamma Q_{\bar\theta}(s_{t+2}^\tau, \pi_{\bar\psi}(s_{t+2}^\tau, g), g) - \delta_{\bar\theta}(s_{t+1}^\tau, a_{t+1}^\tau, g) \right] - \gamma A_{\bar\theta}(s_{t+1}^\tau, a_{t+1}^\tau, g) \\
&\quad - \delta_\theta(s, a, g) && \text{(A11)} \\
&= \cdots \\
&= \sum_{i=0}^{n-1} \gamma^i r_{t+i} + \gamma^n Q_{\bar\theta}(s_{t+i}^\tau, \pi_{\bar\psi}(s_{t+1}^\tau, g), g) - \sum_{i=1}^{n-1} \gamma^i [A_{\bar\theta}(s_{t+1}^\tau, a_{t+i}^\tau, g) + \delta_{\bar\theta}(s_{t+1}^\tau, a_{t+i}^\tau, g)] \\
&\quad - \delta_\theta(s, a, g) \\
&= y^{(n)}(t, \tau, g) - \sum_{i=1}^{n-1} \gamma^i \left[ A_{\bar\theta}(s_{t+1}^\tau, a_{t+i}^\tau, g) + \delta_{\bar\theta}(s_{t+1}^\tau, a_{t+i}^\tau, g) \right] - \delta_\theta(s, a, g). && \text{(A12)}
\end{aligned}
$$

In Eq. (A10), the highlighted term, $\gamma Q_{\bar\theta}(s_{t+1}^\tau, a_{t+1}^\tau, g)$, is purposefully added and then subtracted by $Q_{\bar\theta}(s_{t+1}^\tau, \pi_{\bar\psi}(s_{t+1}^\tau, g), g)$, resulting in $\gamma A_{\bar\theta}(s_{t+1}^\tau, a_{t+1}^\tau, g)$, thereby initiating the unrolling process. Following this, Eq. (A11) showcases the expanded form of the previous term, elucidating the unwinding of the action-value function's recursive definition. This expansion includes the subsequent reward $r_{t+1}$, the ensuing action-value function $Q_{\bar\theta}(s_{t+2}^\tau, \pi_{\bar\psi}(s_{t+2}^\tau, g), g)$, and the TD error $\delta_{\bar\theta}(s_{t+1}^\tau, a_{t+1}^\tau, g)$. Consequently, Eq.(3) from the main text is directly extrapolated from Eq. (A12).

## Appendix B:   Bias Resilient Algorithm

In light of the technical solutions elaborated in the main text, this work culminates in the creation of an advanced algorithm named *Bias-Resilient MHER* (BR-MHER). This innovation stands as a systematic approach to significantly mitigate the challenges posed by both shooting bias and shifting bias. A concise summary of the BR-MHER algorithm is thoughtfully provided in Algorithm 1 for comprehensive understanding and implementation.

The incorporation of MHER($\lambda$) within the framework ensures that the target value aligns as a truncated version of $\hat{y}_\lambda^{(n)}(t, \tau, g)$, a detailed discussion of which is available in Sect. A.4. This target value is mathematically represented as:

$$
\hat{y}_\lambda^{(n)}(t, \tau, g) = \frac{\sum_{i=1}^n \lambda^i \hat{y}^{(i)}(t, \tau, g)}{\sum_{i=1}^n \lambda^i}. \tag{A13}
$$

In this equation, $\hat{y}^{(i)}$ denotes the truncated multi-step targets, as cross-referenced in Sect. 5.2 in the main text. It is imperative to note the presentation of the BR-MHER algorithm in a generalized form, ensuring its broad applicability to the actor-critic framework. This generalization makes it a suitable and effective tool for algorithms such as TD3 (Fujimoto et al., 2018) and DDPG (Lillicrap et al., 2015), reinforcing its utility and versatility in diverse contexts.

## Appendix C:   Experimental and Implementation Details

In this appendix, we furnish extensive details regarding our experimental tasks, implementation specifics, hardware requisites of our experiments, and information pertinent to our released code.

### C.1   Environments and Tasks

In this section, we provide comprehensive elaboration on the four unique task types incorporated into our experimental procedures.

---

**Algorithm 1** Bias Resilient MHER (BR-MHER)

---

**Require:** Environment $E$, Replay buffer $\mathcal{B}$, mini-batch size $b$, reward function $r$, hindsight relabel proportion $p_h$, policy $\pi_\psi$, the quantile $\rho$, Huber loss threshold $\kappa$, action-value functions $Q_\theta^\rho$.

1: **for** $episode = 1, \ldots, N$ **do**
2:   Reset $E$ to get initial state $s_0$ and desired goal $g$.
3:   **for** $t = 0, \ldots, T-1$ **do**
4:     Execute action $a_t \sim \pi(s_t, g)$ and observe $s_{t+1}$.
5:     Store transition $(s_t, a_t, s_{t+1}, g)$ in $\mathcal{B}$.
6:     Update $s_t \leftarrow s_{t+1}$.
7:   **end for**
8:   Sample a batch $B = \{(s_{t_i}^{\tau_i}, a_{t_i}^{\tau_i}, : s_{t_i+n}^{\tau_i}), g_{\tau_i}\}_{i=1}^b$ from $\mathcal{B}$.
9:   Perform hindsight relabeling with $g_{\tau_i} \leftarrow \hat{g}_{\tau_i}, i = 1 \ldots b$ at probability $p_h$.
10:   Calculate target $\hat{y}_\lambda^{(n)}(t_i, \tau_i, g_{\tau_i})$ by Eq. (A13).
11:   Minimize the expected quantile regression loss $\mathcal{L}_{QR}$ (defined in Eq. (7) in the main text) between the target values and the action-value predictions to update $\theta$.
$$L_{critic} = \tfrac{1}{b} \sum_{i=1}^b \mathcal{L}_{QR}(\hat{y}_\lambda^{(n)}(t_i, \tau_i, g_{\tau_i}), Q_\theta^\rho(s_{t_i}^{\tau_i}, a_{t_i}^{\tau_i}, g_{\tau_i}); \rho, \kappa).$$
12:   Maximize the expected actor loss to update $\pi_\psi$.
$$L_{actor} = \tfrac{1}{b} \sum_{i=1}^b Q_\theta(s_{t_i}^{\tau_i}, \pi_\psi(s_{t_i}^{\tau_i}, g_{\tau_i}), g_{\tau_i}).$$
13: **end for**

---

**Fetch Tasks**  The Fetch robotic arm Plappert et al. (2018) represents an advanced, seven-degree-of-freedom (7-DoF) robotic arm outfitted with a two-pronged parallel gripper. The goals are denoted by a 3-dimensional vector that specifies the Cartesian coordinates of the target endpoint. A goal is deemed successfully achieved if the achieved goal resides within a 5cm radius of the intended coordinates. Notably, the arm features a four-dimensional continuous action, which encompasses three Cartesian dimensions to regulate the end effector's movement, along with an additional dimension dedicated to controlling the gripper. For the experimental tasks utilized for our study: i) `FetchPickAndPlace`: The agent is required to grasp the target object, relocate it to the goal position and sustain its position until the termination of the task; ii) `FetchSlide`: The task requires the agent to slide a small puck across the table to a goal position that is outside of the reach of the robot, and ensure the object remains within the goal location at the end of the task.

**Hand Tasks**  The anthropomorphic robotic Plappert et al. (2018) hand is a 24-DoF robot hand with a 20-dimensional action space. For the tasks used in our experiments: i) `HandReach`: The goal of the task is a 15-dimensional vector corresponding to the desired Cartesian positions of all five fingertips. The agent is required to manipulate the robot to reach the states where the mean distance between five fingertips and the goal positions is less than 1cm. Its state is a 63-dimensional vector; ii) `HandManipulateBlockRotateXYZ`: The agent is required to manipulate a block at the palm of the robotic hand to achieve a target pose. The goal is a 7-dimensional vector including 3-dimensional target positions and 4-dimensional target rotations. The agent succeeds in this task if the block differs from the target rotation within 0.1 rad. Its state space is 61-dimensional.

**Sawyer Tasks**  The Sawyer robotic arm Nair et al. (2018) is a 7-DoF robotic arm. Its end-effector (EE) is constrained to a 2-dimensional rectangle parallel to a table. The movements and positioning of this EE are dictated by actions relayed through a motion capture (mocap) system. For the tasks used in our experimental study: i) `SawyerPushAndReachEnvEasy`: This task necessitates the agent to push a small puck on the table to a target XY position specified by the 2-dimensional goal. The task operates within a 2-dimensional action space corresponding to the X and Y axes. The state space for this task, encapsulated in five dimensions, encompasses both the 3-dimensional coordinates of the EE and the 2-dimensional positioning of the puck; ii) `SawyerReachXYZEnv`: This task demands precise manipulation of the robotic arm's EE towards a designated position indicated by Cartesian coordinates. Both the state and the goal spaces are 3-dimensional, encompassing the current and target coordinates of the EE. The action associated with this task is a 3-dimensional continuous variable, each representing the velocity along an axis.

**Grid-World Task** The grid-world tasks employed in this study have been adapted from Chevalier-Boisvert et al. (2018) to suit GCRL contexts. In these tasks, the current state is defined by the agent's current 2-dimensional coordinates, while the goal corresponds to the desired coordinates the agent aims to reach. For these grid-world tasks, we set the agent's action space to consist of five discrete actions: upward movement, downward movement, leftward movement, rightward movement, and maintaining the current position. The tasks `Simple-MiniGrid-Empty-25×25` and `Simple-MiniGrid-Empty-50×50` are differentiated solely by scale, containing 25 and 50 grids along each edge respectively.

The horizon $T$ varies based on the specific task at hand. For instance, the horizon for tasks such as `FetchPickAndPlace`, `FetchSlide`, and `HandReach` is set at 50. In contrast, the tasks `HandManipulateBlockRotateXYZ`, `SawyerPushAndReachEnvEasy`, and `SawyerReachXYZEnv` have a larger horizon of 100. In the context of grid-world tasks, the horizon is determined as three times the scale of their respective scales. Consequently, this results in a horizon of 75 for the task `Simple-MiniGrid-Empty-25×25`, and extends to 150 for the task `Simple-MiniGrid-Empty-50×50`.

## C.2 Implementation Details

In this section, we detail the implementation of our TD3, GCRL, and WGCSL methods, outlining the network configurations and hyperparameters used.

**TD3** TD3 serves as the primary framework for each GCRL method in our experiments, utilizing CDQ, target networks and delayed actor updates as decribed in Sect. A.1 to ensure the stability of the learning process. The hyper-parameters employed in the experiments are shown in Table 1. Notably, in experiments on grid-world tasks, which operate within a discrete action space, we employ the GumbelSoftMax activation (Jang et al., 2016) to the policy network outputs. This approach ensures the action-value function's differentiability relative to the actor outputs within these contexts. Observations and goals, serving as inputs to the actor and critic, are normalized based on historical statistics. Throughout the training phase, the target networks undergo soft updates with the parameters of the present actor and critic networks. Each epoch contains varying numbers of cycles depending on the task: Fetch and Hand tasks comprise 50 cycles per epoch, while Sawyer and grid-world tasks include 10. For exploration, we apply $\epsilon$-greedy strategies and additionally integrate Gaussian action noises into the training rollouts of robotic tasks.

**GCRL Methods** The primary distinctions among GCRL methods stem from the calculation of target values for the critic, as depicted in Section A. All GCRL methods share the same relabeling probability of $p_h = 0.8$, implying an average relabeling of four out of five data entries. For MMHER, we set $\alpha = 0.5$ in Eq. (A6) and utilize the configurations are outlined in Table 2, following the suggestions from Yang et al. (2021). For both IS-MHER based on Retrace($\lambda$) and MHER($\lambda$), we assign a value of 0.7 to $\lambda$ in both Eq. (A9) and Eq.(A5). In IS-MHER where the importance sampling necessiates a stochastic policy, we establish the stochstic policy via adding a clipped Gaussian distibuted noises $\epsilon_a \sim clip(\mathcal{N}(0, 0.2), -0.5, 0.5)$ to the actions. Each experiment is conducted with five distinct random seeds—$\{111, 222, 333, 444, 555\}$—and the statistics of the results are reported.

**WGCSL** Given the focus of WGCSL on supervised learning, we opt not to integrate TD3 into its framework. Our experimental setup is based on the source code provided by the author (Yang et al., 2022). The $K$ used for the weight DRW in Eq. (A8) is set to 10. Apart from the non-application of TD3-specific parameters and the substitution of Softmax for GumbelSoftmax in grid-world tasks, all other hyperparameters are consistent with those utilized in other GCRL methods, see Table 1.

In general, we primarily adhere to the hyper-parameters proposed in the codebase (Yang et al., 2021; 2022) to maintain consistency and avoid parameter tuning. This decision stems from our central aim: to determine if our methodologies can surpass both baseline and contemporary state-of-the-art methods when subject to identical conditions. Importantly, our strategies are tailored to tackle the root causes of bias issues, with the intention to elevate performance through an intrinsic comprehension of these challenges.

Table 1: The hyper-parameters used for TD3

| Hyper-parameter | Value |
|---|---|
| Actor learning rate | 0.001 |
| Critic learning rate | 0.001 |
| Optimizer | Adam (Kingma & Ba, 2014) |
| Buffer size | $10^6$ |
| Quadratic penalty on actions | 1.0 |
| Observation clip range | [-200, 200] |
| Normalized observation clip range | [-5, 5] |
| Random init episodes | 100 (500 for MMHER) |
| Batch size | 1024 |
| Test episodes per epoch | 120 |
| Discount factor $\gamma$ | $1/T$ |
| $\epsilon$ for $\epsilon$-greedy exploration | 0.3 |
| Gaussian action noise std | 0.2 (robotics) |
| Relabeling probability $p_h$ | 0.8 |
| Episodes per cycle | 12 |
| Training batches per cycle | 40 |
| Target network update proportion | 0.005 |
| Delayed actor update interval $k$ | 2 |
| Number of hidden layers | 3 |
| Hidden layer size | 256 (robotics), 512 (grid) |
| Hidden layer activation | ReLU |
| Actor output activation | Tanh (robotics) |
|  | Softmax + GumbelSoftmax (grid-world) |
| Training cycles per epoch | 50 (Hand and Fetch tasks) |
|  | 10 (Sawyer and grid-world tasks) |

Table 2: The hyper-parameters of the world-model used for MMHER

| Hyper-parameter | Value |
|---|---|
| Learning rate | 0.001 |
| Optimizer | Adam |
| Number of hidden layers | 8 |
| Hidden layer size | 256 |
| Batch size | 512 |
| Warmup training times | 500 |

## C.3 HARDWARE

In conducting each experiment, we utilized the processing power of a V100 GPU coupled with eight CPU cores, facilitating efficient model training and comprehensive data collection from environments.

## C.4 CODE

We have adopted a modified OpenAI Baselines codebase (Dhariwal et al., 2017), adhering to the guidelines put forth by Yang et al. (2021; 2022). Tensorflow (Abadi et al., 2016) serves as our chosen deep learning framework for training. Recognizing the pivotal role of reproducibility in scientific endeavors, we have rendered our code publicly accessible. Detailed instructions to aid in the effective usage of the code are provided. This information can be accessed at https://github.com/BR-MHER/BR-MHER.

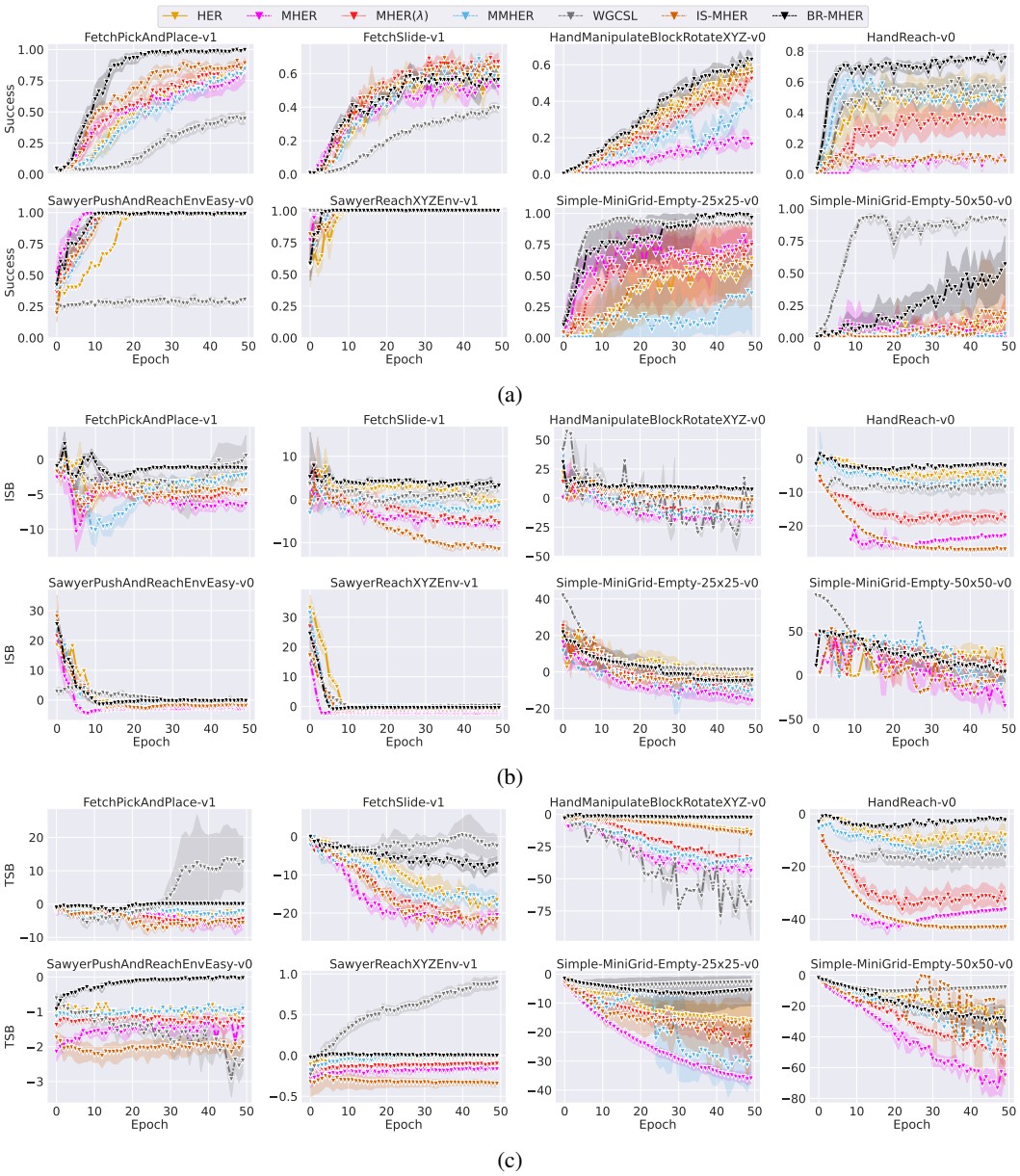

Figure I: Comparative study of three-step ($n$=3) GCRL methods on both robotic tasks and grid-world tasks. (a) Success rate. (b) ISB for measuring shooting bias. (c) TSB for measuring shifting bias.

## Appendix D:    Extended Experiments

In this appendix, we first enrich the comparative analysis of the BR-MHER application, initially presented in the main text, by conducting a deeper examination of both the three-step ($n$=3) and five-step ($n$=5) GCRL methods applied to the same set of tasks. Following this, we conduct an ablation study to elucidate the components within BR-MHER.

### D.1    MORE RESULTS OF BR-MHER

This section delineates the remaining comparative analysis of BR-MHER for $n$=3 and $n$=5, as illustrated in Figs. I and II.

The robust advantages delineated in the main text persist for smaller step sizes such as 3 or 5. It is observed that BR-MHER consistently mitigates off-policy bias, as evidenced by the ISB and TSB levels in Figs. I(b, c) and II(b,c). Here, we can investigate a new research question following the three in the main text: **Q4)** How does the performance of BR-MHER change as the step-size of multi-step GCRl changes? How is it compared to other GCRL method? Instead of performing the one-to-one comparison as the main text, we answer in a general perspective.

As depicted in Figs. I(a) and II(a), BR-MHER overtakes the learning efficiency of HER in `HandManipulateBlockRotateXYZ-v0`, although it aligns with HER's efficiency in a seven-step learning context. This trend highlights that BR-MHER is similarly impacted by the detrimental effects of off-policy biases with increasing step-size. Concurrently, BR-MHER reaps greater advantages than other GCRL methods in grid-world tasks with step-size augmentation. No significant performance shifts are noted for other tasks. An analysis of the performance differential among various multi-step GCRL methods reveals expanding gaps in `FetchPickAndPlace-v1`, Hand tasks, and grid-world tasks. This observation corroborates the enhanced robustness and resilience of BR-MHER to alterations in step-size and escalating off-policy biases.

Moreover, we find the performance does not decrease monotonically as the step-size increases. Each multi-step GCRL method seems to have the best step-size for learning. Take the `Simple-MiniGrid-Empty-25x25-v0` for example, MHER($\lambda$) reaches around 75% success rates with three-step learning, arising to 85% with five-step while dropping to 40% with ten-step learning. In contrst, BR-MHER can stably reach 100%. In this case, BR-MHER offers an robust option to the parameter of step-size. We also notice that there is an special case in `HandManipulateBlockRotateXYZ-v0` that IS-MHER keeps the same performance as HER when $n$ =3, 5 and 7. It probably reduce the multi-step learning to single-step learning via importance sampling due to the relative low entropy of the policy.

Furthermore, the analysis discerns a non-monotonic performance degradation with step-size amplification. Each multi-step GCRL method appears to have an optimal step-size for learning. For instance, in the `Simple-MiniGrid-Empty-25x25-v0` task, MHER($\lambda$) records approximately 75% success with three-step learning, which escalates to 85% with five-step learning, but falls to 40% with ten-step learning. Contrarily, BR-MHER consistently achieves a 100% success rate, underscoring its robustness relative to step-size parameterization. An anomalous observation in `HandManipulateBlockRotateXYZ-v0` shows that IS-MHER mirrors HER's performance for $n$ =3, 5, and 7, potentially signifying a transition from multi-step to single-step learning via importance sampling, attributed to the relatively low entropy of the policy.

### D.2    ABLATION STUDY

This ablation study is systematically designed to unravel the unique contributions of each component embedded within the BR-MHER algorithm. It chiefly emphasizes their roles in the reduction of bias and enhancement of learning efficiency.

As delineated in the primary text, quantile regression and truncated multi-step targets emerge as pivotal techniques, seamlessly integrated within the BR-MHER algorithm. To discern the solitary impact of each, the study explores two algorithmic variants: QR-MHER and TMHER($\lambda$). QR-MHER singularly integrates quantile regression, while TMHER($\lambda$) exclusively embodies truncated multi-step targets, each within the foundational MHER($\lambda$) framework. This methical investigation aims to

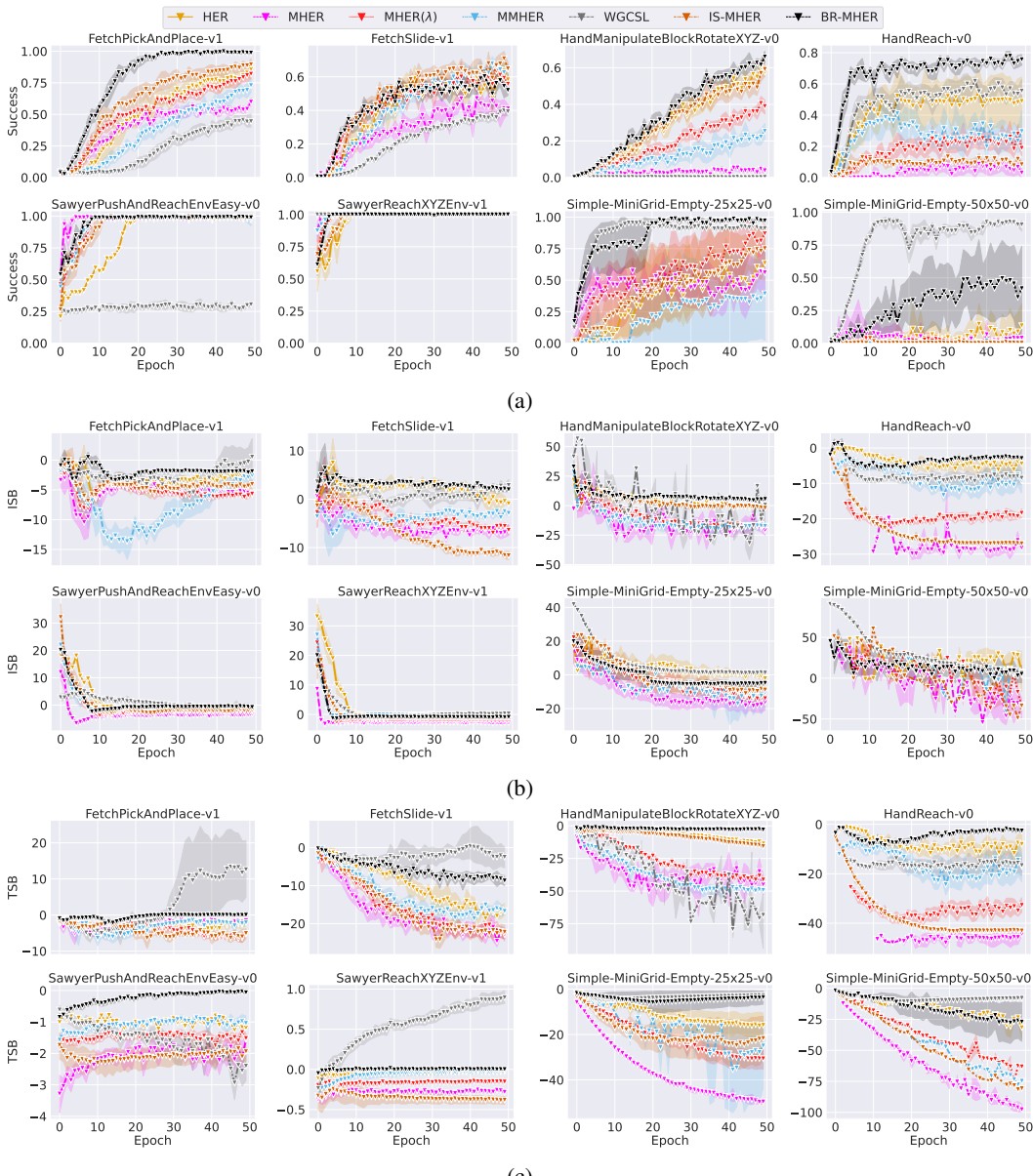

Figure II: Comparative study of five-step ($n$=5) GCRL methods on both robotic tasks and grid-world tasks. (a) Success rate. (b) ISB for measuring shooting bias. (c) TSB for measuring shifting bias.

shed light on the nuanced influences each component imparts on the overarching performance of the BR-MHER algorithm.

Advancing into the ablation study, six methods, namely HER, MHER, MHER($\lambda$), TMHER($\lambda$), QR-MHER, and BR-MHER, are meticulously analyzed. The discerned outcomes are pictorially represented in Figs. III-V.

From Figs. III-V(a), it is evident that MHER($\lambda$) consistently enhances the stability of MHER across diverse tasks and step sizes, thereby providing a more stable performance trajectory.

Our variant, TMHER($\lambda$), stands as a further consistent enhancement of MHER($\lambda$), universally exhibiting positive advancements in success performance, learning efficiency, and off-policy bias mitigation over MHER($\lambda$) across all cases. This underscores the favorable impact of employing truncated multi-step targets. As delineated in Figs. III-V(a), significant improvements with

TMHER($\lambda$) are notably observed in three tasks: `HandReach-v0` and two grid-world tasks. In these scenarios, actions can inadvertently deflect the agent from goal coordinates, culminating in negative rewards in the bootstrap path and an amplified shifting bias. The rectification of this issue leads to substantial enhancement in task performance. It is imperative to note that TMHER($\lambda$) outperforms MHER($\lambda$) in `FetchSlide-v1`, a task where the comprehensive integration of BR-MHER falls short. This observation highlights the potential detrimental impact of quantile regression on the performance within the `FetchSlide-v1` context. Concurrently, despite the improvements in off-policy biases with TMHER($\lambda$) over MHER($\lambda$), Figs. III-V(b,c) reveal that off-policy biases, as indicated by ISB and TSB, maintain high levels in tasks such as `FetchSlide-v1` and `HandManipulateBlockRotateXYZ-v0`. This highlights the circumscribed capacity of TMHER in diminishing both shooting and shifting biases.

In the examination of QR-MHER's performance as depicted in Figs. III-V(a), QR-MHER is observed to markedly excel over TMHER($\lambda$) within the contexts of `FetchPickAndPlace-v1` and `HandManipulateBlockRotateXYZ-v0`. This superiority highlights the advantageous employment of beneficial off-policy bias. Observations from Figs. III-V(b,c) denote that QR-MHER markedly diminishes the off-policy biases of MHER($\lambda$) across all tasks, with the exception of `HandReach-v0` and grid-world tasks. In these exceptional tasks, TMHER($\lambda$) demonstrates superior bias mitigation capabilities. This observed pattern indicates a possible constraint within QR-MHER, specifically in its strategic capacity to exclude bootstrap paths encompassing detrimental actions. This restriction notably leads to a swift accrual of shifting bias. Despite the evidence in Fig. III(a) demonstrating QR-MHER's augmented learning efficiency relative to TMHER($\lambda$) in grid-world tasks when $n$ is 3, this efficiency superiority is overshadowed by TMHER($\lambda$) as $n$ augments. This shift underscores the intensified source of bias within QR-MHER.

Consequently, our BR-MHER amalgamates the advantages of both QR-MHER and TMHER($\lambda$) seamlessly. This integration leads to a superior performance that transcends the peak efficiencies of both QR-MHER and TMHER($\lambda$). In `HandReach-v0` and grid-world tasks, as depicted in Figs. III-V(a), BR-MHER demonstrates an augmented performance compared to the two aforementioned variants. Additionally, the magnitudes of both ISB and TSB are notably reduced, indicative of enhanced off-policy bias mitigation, as evident in Figs. III-V(b,c). Notably, in these tasks where TMHER($\lambda$) surpasses QR-MHER, BR-MHER further diminishes both off-policy biases. This highlights the effective management of both shifting and shooting biases by quantile regression when paired with truncated multi-step targets. Despite a noted performance decline in `FetchSlide-v1` due to quantile regression, BR-MHER maintains robust performance across a variety of tasks and multi-step learning step sizes.

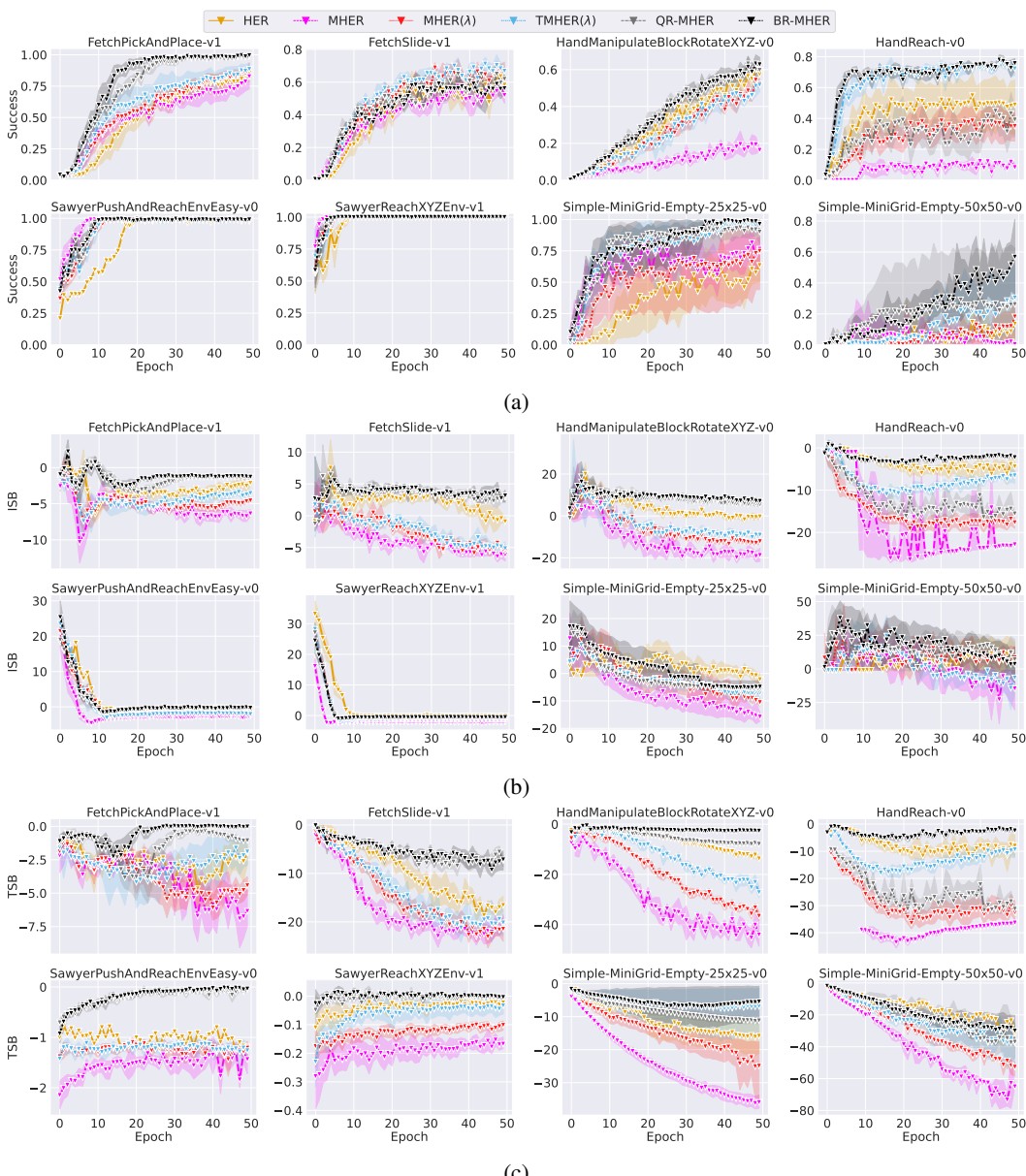

Figure III: Ablation study of three-step ($n$=3) GCRL methods on both robotic tasks and grid-world tasks. (a) Success rate. (b) ISB for measuring shooting bias. (c) TSB for measuring shifting bias.