# OpenReview forum: "Bias Resilient Multi-Step Off-Policy Goal-Conditioned Reinforcement Learning"
_ICLR.cc/2024/Conference — Submitted to ICLR 2024_

### Official Review · Reviewer_GrEH · 2023-10-29

**Soundness:** 2 fair
**Presentation:** 1 poor
**Contribution:** 2 fair
**Rating:** 3
**Confidence:** 4

**Summary:**

This paper studies off-policy biases in multi-step goal-conditioned reinforcement learning (GCRL). It focuses on successful trajectories and divides the biases into two types: shooting bias and shifting bias, which correspond to TD errors of multiple steps and the errors of the action-value function Q at goal states. For the former, the paper suggests that sometimes this bias is helpful and proposes to use quantile regression to utilize this type of bias. For the latter, the paper proposes to truncate the multi-step target to the step in which the goal is first reached. An empirical study shows the proposed algorithm, BR-MHER, consistently performs well on the six tested domains.

**Strengths:**

This paper has a very clear structure and is well organized. It first gives an introduction to GCRL and different types of biases. Then, it provides a literature review on multi-step off-policy (goal-conditioned) RL and quantile-based regression, which is used to utilize potentially useful shifting bias. It then analyzes the multi-step target in GCRL and introduces two types of biases, followed by corresponding treatment. Finally, metric and empirical results are given to illustrate the effectiveness of the improved algorithm. Overall, the logic is clear.

**Weaknesses:**

The major weaknesses are its presentation and soundness.

For the presentation, the paper needs to improve its preciseness and rigorousness. First of all, there are a lot of typos and undefined symbols, which make it difficult to understand the accurate message intended to be delivered (see Questions for an unexhausted list). Apart from these, the paper is also not precise and rigorous in its exposition. For example, 1) in Eq. (1), it defines the policy as the greedy policy of the $Q_\theta$ function, while it is actually a parameterized policy and may be different from the greedy policy; 2) the reuse of $\delta_\theta$ in Eq. (3) is really confusing as it was defined as the TD error already; 3) In Eq. (4) and Eq. (5),  a) $r_i^{\tau^g_\pi}$ is not defined; b) the exponent of the $\gamma$ in the final term is off; c) does Eq. (4) hold? If yes, how?

For its soundness, the biases that the paper proposed are either not precise or not well justified. Despite the effort to isolate the defined off-policy biases from other biases (like over-optimistic bias and hindsight bias), the shooting bias seems to couple with the bias from the target network. As for the shifting bias, it appears that it is just an estimation error instead of a bias due to the off-policy issue of HER. At least from its current presentation, it seems farfetched to call the defined shooting and shifting biases off-policy biases.

**Questions:**

Questions that may affect the assessment:
1. What doesn’t an episode or an n-step trajectory terminate upon reaching a goal? And why can’t we just set the values of goal-reaching states to zero?
2. How can we get Eq. (4)? Isn’t $Q_\theta$ the estimation of the action value function?

Clarity and typos (that have a low impact on the assessment independently):
1. The naming of the algorithms is a bit confusing. In the appendix, all the three variants of the proposed algorithm, TMHER($\lambda$), QR-MHER, and BR-MHER, are using the $\lambda$ target for learning, which is only indicated in TMHER($\lambda$). This may cause confusion that QR-MHER and BR-MHER don’t use the $\lambda$ target.
2. The paper focuses exclusively on deterministic environments, which is mentioned in the introduction but should also be explicitly stated in the preliminary section. Defining $\mathcal{T}$ to be a deterministic transition function will help make this clear.
3. In Eq (1) on page 3, what is $\mathcal{B}$? Is it a dataset?
4. Before section 4 on page 3, “can truncate” should be “can be truncated.”
5. In the first equation (unlabeled) on page 3, 1) $\pi_\phi$ should be $\pi_\psi$; 2) looks like $\theta$ and $\bar{\theta}$ are swapped, which is really confusing.
6. On page 3, the details of the derivation of Eq. (2) are not in Appendix A, which I assume is mistakenly written as the derivation of Eq. (3) in A.8.
7. Several typos in the derivation in A.8: 1) Should the $t+1$s on the left-hand side be $t$? 2) What are $s$ and $a$ in the first equation? 3) Are the subscripts for all $s_{t+1}^\tau$s off? 4) What’s the definition of $r_t$?
8. Be consistent with the naming of the bias. For example, in section 5 on page 4, “shift” should be “shifting,” and “shoot” should be “shooting.”
9. On page 5, “Quatile” should be “Quantile.”
10. On page 6, there is no Eq. (8).

---

### Official Review · Reviewer_NQk6 · 2023-10-31

**Soundness:** 2 fair
**Presentation:** 1 poor
**Contribution:** 2 fair
**Rating:** 3
**Confidence:** 3

**Summary:**

This paper provides an analysis of the bias in goal-conditioned reinforcement learning (GCRL) when using multi-step/n-step targets in temporal difference (TD) learning. In particular, it decomposes the bias in two types: 1) Shooting bias, defined by the sum of the  temporal difference errors in the n-steps along the sampled data trajectory. 2) Shifting bias, defined by the agent's advantage computed on the same n state-action tuples. Hence, to mitigate underestimation from the shifting bias without severely hindering its positive effects for faster reward propagation, they propose to employ a Quantile huber loss, placing a higher weight on the TD-targets with high Q-values, together with other complementary methods from prior work.  Furthermore, to mitigate the shifting bias, they propose to truncate the rewards in the TD-targets whenever a goal-state is reached (which in GCRL, with rewards being uniformly negative outside the goal states, always provides an upper bound on the TD-targets). Empirically, they show gains over prior GCRL algorithms with different values of n.

**Strengths:**

- GCRL and dealing with sparse is a relevant problem where progress can have a concrete impact for several applications of RL.

- I found the formal classification of shifting/shooting bias to be quite interesting.

- The introduced methodology is quite intuitive and follows nicely from the analytical considerations.

**Weaknesses:**

1. I found that the current way the analysis is presented makes it unnecessarily convoluted and hard to read. In particular, I believe there is an overuse of equations without proper context. For instance, Section 5.1 tries to formalize the intuitive argument that for the transitions with high Q-values, the n-step bias is less detrimental and can even aid faster reward propagation, but I feel this argument is never made explicit. I would start each subsection with a statement about what that the authors are trying to claim or about the introduced methodology to improve clarity. Furthermore, some statements also seem imprecise e.g. In connection to the proposed truncated multi-step targets, the authors state "Since the agent is already in the goal states, the need for rapid reward propagation through multi-step learning is naturally reduced." (page 6) However, I believe that in the context of goal-based RL (with rewards always negative for states but the goal) the proposed truncation should always lead to even faster reward propagation since it will always produce an upper-bound to the original targets.

2. I found the empirical results from to be unsatisfactory. The proposed algorithm combines multiple practices and existing methodology to combat two identified sources of bias. However, the paper is entirely lacking ablation studies/analytical experiments to understand which type of bias is most damaging and which component of the proposed methodology is playing the most effect in bias mitigation and performance. For example, is it the huber component of the loss/the quantile component/ or MHER(λ) that is most helping to fend off shooting bias? Furthermore, both the number of environments and training seeds is quite limited.

3. I also found the presentation of the current result to currently be not up-to-standard. For instance, the proposed algorithm is compared with each baseline individually, which I found this to be unnecessarily redundant, taking up a lot in the main text that could be used to better explain the methodology or provide additional results. I believe it would also help to summarize the results and show the actual gains of the proposed algorithm comparing the performance when picking the best value of n for each baseline.

4. I find the experiments to not be fully reproducible. For instance, no details are provided as to how the OpenAI environments are modified (as stated in the 3rd paragraph of Section 6.2), and there are no details/pseudocode descriptions to understand exactly how the TSB and ISB metrics to estimate bias are computed in practice.

Minor:
A few sentences in the text contains grammar errors and repetitions (e.g., the second sentence of page 3, 'Quatile Regression' page 5, 'the Huber loss threshold (κ) is 10.' is stated twice in Section 6.2...). To improve readability, I would suggest passing the paper through a grammar-checker.

**Questions:**

- I did not understand why the authors claim that "the shifting bias is unique to multi-step GCRL," if I am not miss-understanding something, I do not see why the decomposition of Equation 6 would not hold in arbitrary RL problems. Can the authors clarify this point?

- Can the authors explain exactly how the OpenAI codebase is modified? (as stated in the 3rd paragraph of Section 6.2) I think this would be important to clarify for reproducibility, especially since the code is not shared.

While I believe this work has a good potential, I believe it would really benefit from additional empirical analysis, to show the consequences of the shifting/shooting biases. Moreover, I found the overall presentation and experiments to be currently insufficient for a high-quality paper. For these reasons, I currently leaning towards rejection, but I am willing to revise my score if the authors manage to properly address my concerns.

---

### Official Review · Reviewer_aQNs · 2023-11-01

**Soundness:** 2 fair
**Presentation:** 1 poor
**Contribution:** 2 fair
**Rating:** 3
**Confidence:** 3

**Summary:**

This paper studies the bias problem in goal-conditioned reinforcement learning (GCRL) setting. The authors first propose to categorize the bias into 2 types, "shooting" and "shifting" biases. A new algorithm Bias-Resilient MHER (BR-MHER) is propsed. Empirical results are provided to show the proposed method is generally better than a number of other baselines on a number of robotic control tasks.

**Strengths:**

**originality**
- the bias decomposition can be an interesting novel analysis

**quality**
- the presentation of the paper is OK

**clarity**
- the paper is mostly clear

**significance**
- the theoretical analysis on the different types of bias can be interesting for the community
- the empirical results show a fairly significant improvement on performance and bias reduction for the proposed method compared to other baselines.

**Weaknesses:**

Writing and presentation:
- The abstract is a bit too vague, very little information is given. In other parts of the paper, it is also often the case that the wording is fancy but vague and makes it hard to understand what the authors are trying to deliver. For example, how exactly do you define resilience? And what exactly is the robotic benchmark you are using? And how do you know a certain bias is beneficial?

Motivation
- it is unclear to me what is motivating the proposed method, and why we need to decompose the bias, there seems to be no motivating examples on why this is a problem. Additionally, after clipped double Q in TD3, there are a large number of other bias reduction techniques proposed for off-policy learning, how does the proposed method compare to these?

The proposed method
- it is unclear what exactly the authors did in the proposed method. I think there should be a focused discussion in the main paper on what exactly the proposed method does, and how it does differently compared to other baselines.

I believe the paper has some interesting results but can benefit from rewriting and re-organizing.

**Questions:**

- Why should we study the 2 different types of bias, what is a motivating example that justifies the need for this analysis and the design of the algorithm?

---

### Official Review · Reviewer_3TCh · 2023-11-02

**Soundness:** 2 fair
**Presentation:** 1 poor
**Contribution:** 2 fair
**Rating:** 3
**Confidence:** 4

**Summary:**

This paper proposes a new multi-step off-policy GCRL method that can ensure a resilient and robust improvement with a large step. Experiment results show the improved performance of the proposed method on several benchmarks

**Strengths:**

This paper aims to address an important problem for multi-step off-policy. The proposed method can enable a larger step size to speed up GCRL.

**Weaknesses:**

The writing of the paper is not satisfactory, making it hard to follow.

- The paper should spend more space in presenting the new algorithm. Section 4.3 is not clear to me.

- I don't understand what you mean by "learning target for a TD error". (before eq.3)

- I don't quite understand Proposition 1: The advantage function should be positive for the optimal action.

- Why the first term in eq. 3 called the off-policy advantage?

- Have you defined $\delta_{\bar \theta}$?

- Why are 10-step learning scenarios important and challenging?  I believe this is a hyperparameter for the algorithm.

- Near eq. 6: why "for a successful .... are zero". I cannot know why.







There are some typical typos in the paper, making the paper less convincing and assessing the correctness of the paper.

- Eq. 1. I don't understand what you mean by $Q_{\theta}=...$. If it's a value function, why can it equal the n-step target value?

- The equation before eq. 2. I believe in the right hand of the equation: the $\theta$ and $\bar {\theta}$ should be exchanged! Otherwise, the derivation in A.8 does not hold.

- Eq. A12. I think $s_{t+1}$ should be $s_{t+i}$?





The experimental result does not show the superiority of the proposed method, according to Figure 1. The proposed method BR-MHER only achieves better performance on 2 of 8 tasks.

**Questions:**

Please see my comments above.

---

### Meta-Review · Area_Chair_hf47 · 2023-12-11

**Metareview:**

All reviewers recommend rejecting this paper. There was also no rebuttal by the authors objecting to points raised by the reviewers. Thus, I am recommending the rejection of this paper.

**Justification For Why Not Higher Score:**

No rebuttal from authors, so no one is fighting for it to be accepted.

**Justification For Why Not Lower Score:**

N/A

---

### Decision · Program_Chairs · 2024-01-16

Reject